# Molecular Mechanisms of Fetal and Neonatal Lupus: A Narrative Review of an Autoimmune Disease Transferal across the Placenta

**DOI:** 10.3390/ijms25105224

**Published:** 2024-05-10

**Authors:** Armando Di Ludovico, Marta Rinaldi, Francesca Mainieri, Stefano Di Michele, Virginia Girlando, Francesca Ciarelli, Saverio La Bella, Francesco Chiarelli, Marina Attanasi, Angela Mauro, Emanuele Bizzi, Antonio Brucato, Luciana Breda

**Affiliations:** 1Paediatric Department, University of Chieti “G. D’Annunzio”, 66100 Chieti, Italy; armandodl@outlook.com (A.D.L.); francesca.mainieri@outlook.it (F.M.); virginiagirlando@gmail.com (V.G.); francescaciarelli89e@gmail.com (F.C.); saveriolabella@outlook.it (S.L.B.); chiarelli@unich.it (F.C.); marina.attanasi@unich.it (M.A.); 2Paediatric Department, Buckinghamshire Healthcare NHS Trust, Aylesbury-Thames Valley Deanery, Aylesbury HP21 8AL, UK; marta.rinaldi@nhs.net; 3Department of Surgical Science, Division of Obstetrics and Gynecology, University of Cagliari, Cittadella Universitaria Blocco I, Asse didattico Medicina P2, Monserrato, 09042 Cagliari, Italy; dr.dimichelestefano@gmail.com; 4Pediatric Rheumatology Unit, Department of Childhood and Developmental Medicine, Fatebenefratelli—Sacco Hospital, Piazzale Principessa Clotilde, 20121 Milan, Italy; 5Division of Internal Medicine, ASST Fatebenefratelli Sacco, Fatebenefratelli Hospital, University of Milan, 20121 Milan, Italy; emanuele.bizzi@asst-fbf-sacco.it (E.B.); antonio.brucato@unimi.it (A.B.); 6Department of Biomedical and Clinical Sciences “Sacco”, University of Milano, Ospedale Fatebenefratelli, 20121 Milan, Italy

**Keywords:** neonatal lupus erythematosus, congenital heart block, autoantibodies

## Abstract

This study, conducted by searching keywords such as “maternal lupus”, “neonatal lupus”, and “congenital heart block” in databases including PubMed and Scopus, provides a detailed narrative review on fetal and neonatal lupus. Autoantibodies like anti-Ro/SSA and anti-La/SSB may cross the placenta and cause complications in neonates, such as congenital heart block (CHB). Management options involve hydroxychloroquine, which is able to counteract some of the adverse events, although the drug needs to be used carefully because of its impact on the QTc interval. Advanced pacing strategies for neonates with CHB, especially in severe forms like hydrops, are also assessed. This review emphasizes the need for interdisciplinary care by rheumatologists, obstetricians, and pediatricians in order to achieve the best maternal and neonatal health in lupus pregnancies. This multidisciplinary approach seeks to improve the outcomes and management of the disease, decreasing the burden on mothers and their infants.

## 1. Introduction

In 1954, Bridge and Foley observed that the maternal “lupus erythematosus factor” can be passed on to the baby. In the same year, a lupus rash was observed in a 6-week-old infant who was born to a mother who was later diagnosed with systemic lupus erythematosus [1]. In 1957, a congenital cardiac disease was identified in an infant whose mother also had the same condition. Subsequently, transient cytopenia and abnormally elevated aminotransferases were reported in infants with neonatal lupus erythematosus [2]. The human fetus does not generate significant amounts of immunoglobulin G and only produces minimal quantities of immunoglobulins A and M. Consequently, the entire collection of antibodies in the fetus and newborn is obtained from the mother. Maternal immunoglobulin G (IgG), but not immunoglobulins A (IgA) or M (IgM), starts to pass through the placenta around the third month of pregnancy. The level of fetal immunoglobulin G gradually increases until it surpasses the maternal level by approximately 10% at term [3]. Fetal or neonatal diseases, such as antiphospholipid antibody syndrome, Graves–Basedow disease, immune thrombocytopenic purpura, myasthenia gravis, neonatal autoimmune blistering disease, Sjögren’s syndrome, and systemic lupus erythematosus (SLE), are caused by versions of immunoglobulin G that target autoantigens. These immunoglobulins can be transferred across the placenta into fetal circulation, leading to significant consequences for the fetus or newborn [4]. Neonatal lupus erythematosus is a peculiar clinical syndrome that affects infants born to women with autoantibodies that act against Sjögren’s syndrome autoantigen types A or B. This article focuses on the molecular mechanisms that lead to neonatal lupus. Although the presence of typical autoantibodies is commonly associated with this condition, it is necessary to consider the potential for other additional factors to also play a significant role in determining the expression of the disease. The management of pregnancy and newborns with congenital heart block is also extensively reviewed [5].

## 2. Molecular Mechanisms

### 2.1. Placenta: Immunological Interface and IgG Transfer

The placenta functions as an immunological interface between the mother and fetus, selectively enabling the transfer of IgGs from the 12th week of pregnancy onwards. IgGs are sufficiently small to cross the placental barrier and reach the fetal bloodstream, thereby offering immunological defense during the early months of the newborn’s life [6]. The neonatal FcR receptor (FcRn) is present on the maternal syncytotrophoblasts of the placental villi. This enables the passage of IgGs via pinocytosis. When IgGs enter the acidic endosome, they are protected from degradation by the lysosome. This enables their release into fetal circulation at a physiological pH [7]. The histopathological changes to the placenta during lupus consist of complement depositions, inflammatory infiltrates, and vasculopathy, which are observed under microscopic examination [8]. These changes not only affect the transport function of IgGs, but may also be involved in the general pathogenesis of lupus by enabling the transfer of autoantibodies that are part of the development of neonatal lupus. Placental pathology reflects the maternal autoimmune response and has consequences for fetal outcomes, highlighting the importance of detailed placental evaluation in lupus pregnancies [8].

### 2.2. Autoantibodies Anti-SSA/Ro and Anti-SSB/La and Their Association with Autoimmune Diseases

As mentioned previously, neonatal lupus is associated with the transfer of anti-SSA/Ro and anti-SSB/La antibodies across the placenta. Anti-nucleus maternal antibodies can be detected in a significant proportion of Sjögren Syndrome patients, ranging from 80% to 90%. These antibodies are also present in other autoimmune diseases like SLE and rheumatoid arthritis, as well as in a small percentage (up to 1.5%) of healthy pregnant women who test positive for anti-SSA/Ro antibodies [9]. Furthermore, the occurrence of fetal complications is associated with elevated levels of antibodies in the maternal bloodstream, rather than simply their presence [10]. Anti-Ro antibodies target four distinct micro-RNAs with molecular weights of 45, 52, 54, and 60 kDa. However, only anti-SSA/Ro52 and anti-SSA/Ro60 are commonly used in clinical assessment.

### 2.3. Fetal Antigens and Immune Complex Formation

Ro52 is an enzyme that has been identified as a ubiquitin E3 ligase, which plays a role in the process of adding ubiquitin molecules to proteins, as well as in pro-inflammatory mechanisms and cell death. Additionally, it is a component of the TRIM21 receptor family, which regulates the activation and proliferation of T-cells and the production of pro-inflammatory interleukins like IL-2. Anti-Ro52 antibodies are commonly associated with increased sensitivity to UV light and congenital heart blocks in neonatal lupus erythematosus (NLE), particularly when targeting the p200 epitope. This specific antibody has been shown to significantly raise the risk of second- and third-degree atrioventricular block [11]. Conversely, Ro60 is an RNA-binding protein that also plays a role in cellular adhesion. Certainly, the increased production and accumulation of Ro60 subsequent to UV exposure or oxidative stress promotes cell survival [12]. Anti-La antibodies specifically target the 48kDa phosphoprotein, which is related to RNA polymerase III. These antibodies are typically detected alongside anti-Ro antibodies. There is no statistical evidence suggesting an increased risk of developing congenital heart block (CHB) in fetuses of mothers who test positive for anti-Ro compared to those who test positive for anti-La. However, it is possible that anti-La is responsible for less than 1% of CHB cases in patients with neonatal lupus erythematosus (NLE). Furthermore, the occurrence of the condition is more prevalent when individuals test positive for both anti-Ro and anti-La antibodies [13,14], and even more so when they test positive for all three antibodies (anti-Ro52, anti-Ro60, and anti-La) [15]. Additionally, these antibodies exhibit the ability to identify cardiac fetal antigens, particularly anti-Ro52. They antagonize the activation of the L-type calcium channels, induced by the presence of serotonin in fetal atrial cells.

### 2.4. Apoptosis, Cardiac Damage, and Tissue Remodeling

Studies conducted in a laboratory setting show that when cardiomyocytes undergo apoptosis, they start expressing Ro and La antigens on their cell surfaces. These antigens are then identified by maternal anti-Ro and anti-La antibodies [16]. Apoptosis plays a critical role in embryogenesis by selectively removing a substantial number of cells within organs. Any disruption to this non-inflammatory process leads to tissue damage. The interaction between the antibody and the antigen (immune complexes) prevents the natural process of programmed cell death and the removal of dying heart muscle cells, leading to the activation of an inflammatory reaction. Research suggests that in vivo opsonization may account for the transition from a non-inflammatory process to an inflammatory one. Specifically, the phagocytosis of opsonized cells is thought to be pro-inflammatory compared to the phagocytosis of apoptotic cells alone [16]. Afterwards, resident macrophages release TNFα and may additionally expose antigen to lymphocytes, thus amplifying the inflammatory response [17,18]. Macrophages also secrete TGF-β, which causes fibroblasts to undergo differentiation into myofibroblasts, ultimately resulting in the formation of scar tissue. Cardiac tissue damage leads to myocarditis, hemorrhage, fibrosis, calcification, and necrosis of the conduction system. Consequently, this process leads to the development of various degrees of heart block, myocardial dysfunction, and/or endocardial fibroelastosis (EFE), ultimately leading to congestive heart failure [19,20]. In summary, maternal anti-Ro and anti-La antibodies pass through the placental barrier and specifically target fetal antigens that are typically expressed during apoptosis. The binding of antibodies to antigens leads to the formation of immune complexes, which triggers the activation of the inflammatory response and consequently leads to damage to fetal tissue, specifically cardiac tissue. Macrophage activation causes fibrosis and permanent damage to the cardiac tissue and its conduction system.

## 3. Congenital Heart Block and Other Complications

### 3.1. Overview

Pregnancies in women who test positive for anti-Ro/SSA antibodies can lead to negative outcomes. These maternal antibodies may cross the placenta, resulting in the fetus displaying autoimmune reactions. The most common and dangerous manifestation is congenital heart block (CHB), which is a form of autoimmune disease that disrupts cardiac conduction tissue [21,22]. As shown in the literature, between 60% and 90% of CHB cases are considered to be related to the passage of maternal antibodies through the placenta to the fetus in the absence of cardiac structural abnormalities [23,24]. CHB occurs in only 2% of fetuses in anti-SSA/RO-positive pregnancies; however, its recurrence is nine times higher (16–18%) in subsequent pregnancies, implying the likelihood that additional maternal, fetal, and environmental factors will impact the immune response and the myocardial fibrosis process [15,25,26,27]. This is further demonstrated by the fact that cardiac involvement may only occur in one of the twins or two of the triplets exposed to maternal anti-Ro/SSA-La/SSB antibodies [28,29]. In the case of both anti-Ro/SSA and anti-La/SSB, the CHB incidence is 3%, and it is even higher in the presence of active disease and/or high-antibody titers [30]. Large prospective studies suggest that most deaths from CHB due to NL occur in utero [31,32,33]. Other cardiac pathological conditions have also been described less frequently [13,34,35,36,37].

### 3.2. Pathogenesis

So far, there is no complete understanding of the precise pathogenetic pathways of CHB. Two main, non-mutually exclusive hypotheses have been advanced that attempt to explain the molecular mechanisms by which the fetal heart is injured by the action of anti-Ro/SSA-La/SSB antibodies. According to the first hypothesis, circulating maternal autoantibodies bind the intracellular target antigens as they translocate to the surface of cardiomyocytes, causing them to undergo apoptosis during physiological myocardial remodeling. The result is thus the inhibition of the normal physiological removal of apoptotic cells and the stimulation of pro-inflammatory and profibrotic reactions in response to the formation of pathogenic antibody–apoptotic immune complexes [38,39,40]. The second hypothesis posits a cross-reaction between these antibodies and the L-type calcium channels, which are surface antigens of the sarcolemma, according to a molecular mimicry mechanism. This process, alone or in combination with the first mechanism, causes the dysregulation of calcium homeostasis and, consequently, cardiomyocytes toxicity [41,42]. Of note, the imbalanced calcium homeostasis is crucial for the propagation of the action potential and for conduction in the atrioventricular (AV) and sinoatrial nodes. Apoptosis and inflammation stimulate macrophages to secrete TGF-β1 and TNF-α. On autopsy, these are found to have permanently high expressions in the conduction tissue of babies with cardiac diseases [31,43]. Fibrotic tissue and calcific regions replace the AV node and surroundings tissues, interfering with the electrical signal [16,32,44,45]. Additionally, examining the electrophysiologic and molecular aspects of CHB, some researchers have demonstrated that anti-Ro/SSA antibodies might directly cause arrhythmogenic activity [38,46]. Over the last few decades, some investigators have attempted to identify possible biomarkers of cardiac damage in these patients. Cardiac damage is thought to be linked to the type and titer of autoantibodies [10]. Two polypeptides, 52 kDa and 60 kDa, constitute the Ro/SSA ribunocleoprotein and serve as targets for anti-Ro52 and anti-Ro60 autoantibodies, respectively [47]. Among these, anti-Ro52 shows a primary function in the initial CHB injury, in particular its central portion (amino acids 200–239), typically referred to as p200. Thus, it is a candidate biomarker that causes an elevated maternal risk for the development of cardiac NL in an offspring [48,49]. However, the combined presence of anti-Ro52, anti-Ro60 and anti-p200 has been more frequently reported in pregnancies with affected fetuses. Therefore, they are good markers, particularly in high concentrations [50]. Conversely, in comparison to Ro52 or Ro60 antibodies, p200 reactivity did not raise the incidence of conduction problems [51]. Therefore, it is currently unclear whether anti-p200 antibodies are actually responsible for CHB pathogenesis, and the same is true for their use as cardiac damage biomarkers [34].

In the context of macrophages that lead to cardiac lesions, sialic acid-binding Ig-like lecithin 1 (SIGLEC-1)-positive macrophages are the most common. These are regulated by type I IFN, which also upregulates Ro52 and induces apoptosis, proving to be a crucial cytokine in the progression of CHB [52,53]. Furthermore, higher levels of SIGLEC-1 and IFN-α have been found in mothers with affected children, rather than in mothers with unaffected children [54]. Both anti-Ro/La-exposed women and their neonates revealed an increased expression of INF-regulating genes with elevated INF-α levels, higher than those found in newborns from healthy mothers. Therefore, maternal IFN is suggested as a novel biomarker for the evaluation of CHB risk [55]. Furthermore, the polymorphism of codon 25 in the TGFB1 gene has been linked to interindividual variability and a high risk of CHB development [56]. Elevated levels of C-reactive protein, N-terminal pro-B-type natriuretic peptide, matrix metalloproteinases (especially type 2), and plasminogen, as well as the presence of urokinase plasminogen activator in umbilical cord blood, have been found in NL patients with extensive cardiac damage. This provides further support for the theory linking these alterations to immune-mediated inflammation and fibrosis [31].

### 3.3. Clinical Presentation

CHB is usually permanent, and its clinical signs are influenced by the ventricular rate. Typically, AVB develops between the 18th and the 26th week of gestation over a short period of about 7 days. However, Makadia et al. [57] showed cases where AVB onset occurred post-24 weeks of gestation. Their study underscored the critical need for vigilant fetal monitoring beyond the conventionally recognized risk period, as delayed manifestation of AVB can have profound implications for perinatal care and outcomes. The severity of conduction disorder varies with the degree of scarring [20]. Earlier stages of AV block (1° or 2°) may respond to anti-inflammatory treatment, but complete (3°) AV block, which can occur within 24 h of a normal fetal rhythm, is likely irreversible and carries substantial risk for significant morbidity and for mortality [58,59]. CHB may occur in both the foetus and neonate; however, it has never been documented in the mother’s heart, even though identical antibodies are present in the mother’s circulation [20]. Advanced second- and third-degree AV block can be detected in utero via the identification of fetal bradycardia. Particularly, the atrial rate is normal, while ventricles generally beat at a rate between 40 and 80/min, and it may also happen that the ventricular rate slows down as the pregnancy progresses [13,34,35,36,49,60,61]. Between the 6th and 8th month of postnatal life, maternal antibodies typically stop being detectable, and this is followed by the regression of hepatic, hematological, and cutaneous symptoms. Conversely, cardiac tissue does not regenerate during this time; thus, most CHB cases are complete and irreversible [62]. After birth, neonates with CHB usually display less than 100 beats per minute [13,34,35,36,49,60,61]. The result of such a low heart rate may be fetal hydrops or neonatal heart failure. Additional extracardiac manifestations are diaphoresis, pallor, peripheral edema, and prominent jugular veins. Some crackles may be present during lung auscultation, while intermittent cannon waves, an initial heart sound with various intensity levels, and intermittent gallops and murmurs may be heard during cardiac auscultation [13,34,35,36,49,60,61]. CHB is linked to an overall mortality that ranges between 9% and 25%, with 70% of deaths occurring in utero [31,32,33]. Other cardiac manifestations have been described in these patients [63].

### 3.4. Diagnosis

The diagnosis of CHB typically occurs during pregnancy, but it can also be identified at birth or within the neonatal period [13]. Given the significant morbidity and mortality associated with congenital heart block (CHB) among newborns, it is highly recommended that the management of infants affected by NLE syndrome be conducted in a tertiary care center that provides comprehensive medical services [9,64,65]. This is crucial as NLE syndrome requires the expertise and collaboration of various subspecialists and the availability of advanced diagnostic tools and technology, enabling the accurate assessment and monitoring of the infants’ condition. The most useful and least invasive diagnostic and surveillance tools for CHB in NLE are fetal echocardiograms (ECHOs), which are used for PR interval measurement [66]. Frequent at-home fetal heart rate monitoring through patient-operated Doppler devices is emerging as a vital component in the management of anti-SSA/Ro-positive pregnancies. Research by Howley et al. [67], alongside Cuneo et al. [68], supports this protocol’s utility, advocating for its integration into routine prenatal care to enable the early detection of cardiac anomalies. This self-monitoring regime could significantly mitigate the risks associated with fetal cardiac arrhythmias. The prolongation of the PR interval to a duration exceeding 150 ms, as determined through echocardiographic monitoring, can be considered to be a valuable “biomarker”, indicative of reversible injury [66]. After the 26th week of gestation, if there is no atrioventricular heart block present, the frequency of echocardiographic monitoring may be reduced [69]. However, it is important to note that less than 20% of cases are detected during the later stages of pregnancy, and so monitoring should still be considered [9]. On the other hand, some studies [21] recommend an earlier screening. This is because, from the 13th week, IgG antibodies are transferred from the mother to the fetus through the placenta. Combined with the timing of the formation of the anatomical function of the cardiac conduction system, damage to the cardiac conduction tissue can occur as early as approximately 8–13 weeks of gestational age [6,70].

### 3.5. Prognosis for Infants with Maternal Anti-RO/SSA Antibodies

The prognosis for infants affected by congenital heart block (CHB) as a secondary condition to maternal anti-RO/SSA antibodies is significantly poor, particularly in the presence of certain adverse factors. These factors include a gestational age at diagnosis of less than 20 weeks, a ventricular rate below 55 beats per minute (bpm), the presence of hydrops fetalis, impaired left ventricular function, cardiomegaly, atrioventricular valve regurgitation, endocardial fibroelastosis, and a low velocity of aortic flow [9,71,72]. The mortality rate in CHB varies between 10 and 29%, and 63–93% of the surviving patients require pace-maker implantation [9,36,61,65,73]. Postnatal pacing strategies include the use of temporary external devices in neonates with severe hydrops or low birth weights, with a transition to permanent internal devices as condition stabilizes [74]. Indications for the implantation of permanent pacemakers in infants include a decrease in ventricular rate to less than 55 bpm, an increase in atrial rate to more than 140 bpm, the presence of wide QRS complexes, congenital heart disease, heart failure, and prolonged QT intervals [75]. The prognosis for these infants can be favorable if there is the preservation of ventricular function. Children diagnosed with NLE have been proved to be at a higher risk of developing autoimmune diseases later in life such as juvenile idiopathic arthritis, psoriasis, thyroid disease such as Hashimoto thyroiditis, iritis, and type 1 diabetes mellitus [72]. Children are also prone to growth retardation and neurodevelopmental problems [76]. Given the familial aggregation of autoimmune diseases [77], a further assessment of the future health of the siblings of children with CHB is recommended.

### 3.6. Management

Given the significant morbidity and mortality associated with CHB among newborns, it is imperative to prioritize early treatment strategies for this condition. While certain studies have proposed the use of antenatal betamethasone or dexamethasone as a preventive measure against CHB, the actual efficacy of such interventions remains unconfirmed [9]. Moreover, the potential toxicity arising from the administration of high-dose corticosteroids throughout pregnancy, particularly in light of the relatively low risk of CHB, is deemed unacceptable [78]. The treatment of the mother and fetus with fluorinated steroids, mainly dexamethasone, may be associated with some deleterious effects on the developing central nervous system, lungs, retina, and adrenal glands [70]. Additionally, it was hypothesized that prophylactic treatment with intravenous immunoglobulin (IVIg) could modulate macrophages and enhance the catabolism of maternal antibodies.

In the quest to mitigate congenital heart block (CHB) in neonatal lupus, the literature showed a spectrum of IVIg dosing protocols. A prospective trial by Friedman et al. [79] administered IVIg at 400 mg/kg triweekly from gestation weeks 12 to 24, a regimen that did not yield a reduction in CHB incidence. Routsias et al. [80] investigated the modulation of the idiotype–anti-idiotype antibody ratio by IVIg without delineating the doses, shifting the focus to immunological mechanisms rather than direct clinical outcomes. Collectively, these investigations reveal the critical necessity of delineating an optimized IVIg dosing regimen to effectively prevent or treat CHB in the context of maternal autoimmunity [79,81,82]. In recent years, there has been growing discussion regarding the potential application of antimalarial drugs, particularly hydroxychloroquine (HCQ). HCQ has been suggested to offer prophylactic benefits with an acceptable level of risk. It is believed that HCQ exerts its effects by blocking the toll-like receptor response, thereby attenuating the immune reaction to self-antigens [83]. It is recommended that HCQ treatment should be started between the 6th and 10th gestational weeks at a dose of 400 mg orally once a day [13,84], and an early treatment with HCQ decreases the recurrence of CHB by >50% [66]. However, cautions use and monitoring in pregnancy are suggested, considering that HCQ may have a role in prolonging the QTc interval in pregnant women, which might extrapolate to fetal effects [85]. To date, this method of prophylaxis is not still recommended in low-risk patients, such as anti-Ro/SSA-positive women without any previous affected child [66].

## 4. Management

During pregnancy, lupus improves in approximately one-third of women, remains stable in another third, and worsens in the remaining third. Therefore, in any particular pregnancy, this clinical condition may deteriorate or undergo a sudden exacerbation without warning [86,87]. Petriet et al. [88] reported a 7% of risk of significant morbidity during pregnancy. In a cohort of 13,555 women with SLE during pregnancy, the maternal mortality and severe morbidity rate was 325 per 100,000 [89]. In a review of 13 studies covering 17 maternal deaths attributable to SLE and lupus nephritis, all occurred in patients with active disease [90]. During the past several decades, pregnancy outcomes in women with SLE have improved remarkably. Pregnancy outcomes are relatively favorable for most women with inactive or mild/moderate SLE. Women who have confined cutaneous lupus do not usually have complications [91]. However, newly diagnosed SLE during pregnancy tends to be severe [92]. In general, pregnancy outcomes are more favorable in women in the following circumstances: (1) lupus activity has been quiescent for at least six months before conception; (2) there is no lupus nephritis manifesting with proteinuria or renal dysfunction; (3) antiphospholipid syndrome or lupus anticoagulant is absent; (4) superimposed preeclampsia does not develop [93,94,95,96]. Active nephritis is associated with adverse pregnancy outcomes, although the prognosis has improved remarkably, especially if the disease remains in remission [94,97].

Regarding complications, women with renal disease have a high incidence of gestational hypertension and preeclampsia. Chronic hypertension complicates up to 30% of pregnancies in women with SLE [98]. Additionally, as mentioned previously, risk of developing preeclampsia is increased, with an incidence rate of 4.3% compared to 0.5% in women without SLE [99]. Preterm birth is the most common obstetric complication in women with SLE [100]. Rates of preterm birth from 15 to 50% are reported, with increased incidence in women with lupus nephritis or high disease activity [100]. Prematurity is a known cause of lifelong impairments such as motor, cognitive, and behavioral problems [101]. Therefore, SLE pregnancies have significant risks and require critical clinical attention. Pregnancy outcomes are better when there is planning and close monitoring of the patient during this process. The American College of Rheumatology (ACR) published evidence-based guidelines, detailing a roadmap to optimize maternal and fetal outcomes in patients with rheumatic diseases [102]. From preconception to parturition, the use of a multidisciplinary team approach with close rheumatologic, obstetric, and neonatal monitoring is critical. Lupus management primarily monitors fetal well-being and maternal clinical and laboratory status [103].

For patients seeking pregnancy, achieving remission or a Low Lupus Disease Activity State (LLDAS) should be the goal before attempting to conceive. Active disease status increases the risk of adverse outcomes for both the mother and the baby. Therefore, to minimize the risk of APOs, patients should maintain LLDAS or remission for six months before attempting conception by using medications compatible with pregnancy. Those with moderate or severe disease activity should delay pregnancy until the disease is controlled using stable, pregnancy-compatible medications [100].

Women with SLE planning pregnancy should undergo testing for the presence of anti-Ro/SSA, anti-La/SSB, and antiphospholipid antibodies (e.g., anticardiolipin IgG, and IgM, b2-glycoprotein-I IgG, and IgM, and lupus anticoagulant (LAC). These antibodies are indicative of potential severe complications during pregnancy. The highest risk for poor fetal and maternal outcomes is associated with moderate and high titers of antiphospholipid antibodies in combination with the LAC (20). Regular assessments during pregnancy are crucial to identifying potential complications and providing early intervention.

Typical symptoms experienced at the beginning of a normal pregnancy encompass fatigue, weight gain, nausea, and increased urinary frequency. During each prenatal visit, the healthcare provider should conduct a physical examination, monitor blood pressure for indications of SLE disease activity, and recommend patients engage in home blood pressure monitoring. An early fetal ultrasound is essential in order to confirm pregnancy localization and establish fetal age [102].

Laboratory assessments, conducted at least once per trimester, should include a complete blood count with differential, double-stranded DNA; measurements of CRP, ESR, C3, C4, anti-dsDNA, and serum uric acid; urinalysis with microscopy; and a urine protein-to-creatinine ratio (UPCR). The initiation of low-dose aspirin (81 mg/day) treatment early in pregnancy is recommended to reduce the risk of preeclampsia [104]. To prevent foetal loss and blood clots, OB-APS patients should receive prophylactic heparin or low-molecular-weight heparin and low-dose aspirin. Anticoagulation should last 6–12 weeks after childbirth. Follow-up appointments depend on the patient’s clinical condition and the physician’s assessment [103]. During pregnancy, it is important to be watchful in order to identify lupus disease activity that may not be apparent through clinical examination but can be detected through regular laboratory tests. Blood tests should be performed regularly to detect active lupus nephritis, preeclampsia, and diabetes [104]. Fetal sonography should estimate the baby’s anatomy and monitor amniotic fluid levels, especially in the second and third trimesters. Doppler sonography of the fetal and uterine arteries is recommended for addressing placenta concerns. Fetal and maternal conditions will determine the delivery method and timing [102].

In summary, systemic lupus erythematosus (SLE) impacts women of reproductive age, requiring family planning and close monitoring. Women with anti-Ro/SSA and anti-La/SSB antibodies require specific monitoring. Prior to conception, it is crucial for patients with systemic lupus erythematosus (SLE) that their condition is well managed with appropriate medications. This is necessary to prevent flare-ups during pregnancy, which may lead to adverse outcomes such as pregnancy loss, pre-eclampsia, premature birth, and small size in gestational-aged infants.

## 5. Conclusions

NLE is a peculiar clinical syndrome that affects infants born to women with autoantibodies against autoantigen types A or B of Sjögren’s syndrome; thus, the management of pregnancy and newborns is crucial and requires critical attention. Over the last few decades, pregnancy outcomes in women with SLE have improved remarkably, although newly diagnosed SLE during pregnancy tends to be severe and characterized by various complications, such as preeclampsia and preterm birth. Nevertheless, women with SLE can have successful pregnancies with better outcomes when the patient care is carefully planned and closely monitored by a multidisciplinary team approach. The presence of anti-Ro and anti-La antibodies is associated with the alteration of physiological cellular apoptosis during embryogenesis. This disruption initiates an inflammatory process that stimulates the formation of fibrotic tissue, directly impacting the myocardium or the conduction system of the newborn. Consequently, a range of cardiac complications can manifest, with CHB being the most frequent and life-threatening manifestation. Therefore, it is highly advisable to seek management in a tertiary care center to ensure the best possible outcomes for infants. Especially with regards to CHB, the severity of this medical condition underscores the importance of this recommendation, as it necessitates expertise and the collaboration of various subspecialists, as well as access to advanced diagnostic tools and technology. ECHO remains the gold standard for diagnosing CHB. It is common practice to perform serial fetal ECHOs on a weekly basis, starting from the 16th week of gestation and continuing until at least the 25th week of gestation. Early treatment strategies for CHB are of the utmost importance as prioritizing early intervention can help to mitigate the morbidity and mortality associated with CHB among newborns. With regards to this, HCQ performs a crucial role as a potential prophylactic option with an acceptable level of risk (management in Figure 1). This review aims to synthesize and present key findings from the existing literature that underscore the criticality of conducting early diagnoses of NLE and implementing a comprehensive cardiovascular follow-up for offspring born to mothers with SLE, starting from the prenatal stage and continuing throughout adolescence. By integrating prenatal screening, routine cardiac assessments, and long-term monitoring, healthcare providers can effectively identify and manage potential cardiac complications associated with NLE, thereby improving the overall prognosis and quality of life for these individuals.

## Figures and Tables

**Figure 1 ijms-25-05224-f001:**
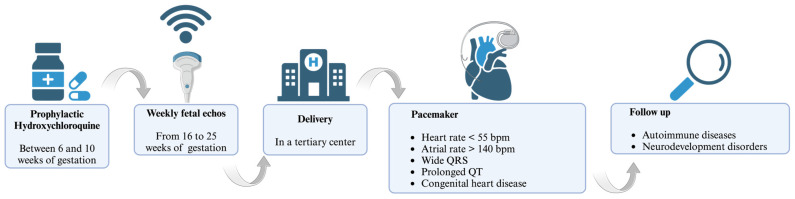
Management of Neonatal Lupus.

## Data Availability

Not applicable to this article as no datasets were generated or analyzed during the current study.

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
