# Peer review of "Molecular Mechanisms of Fetal and Neonatal Lupus: A Narrative Review of an Autoimmune Disease Transferal across the Placenta"

_ijms, 2024, doi:10.3390/ijms25105224_

Round 1

Reviewer 1 Report

Comments and Suggestions for Authors

As reflected in the title, this paper is a narrative review on both maternal and perinatal (fetal and neonatal) lupus.

I'm not sure to what extent the current title covers sufficiently well the maternal aspects as discussed. I hereby noticed that - at present - here are not obstetricians involved in the authorship, and as being a neonatologist myself i cannot fully assess the maternal aspects. Although all maternal management statements read very reasonable, perhaps obstetrics should be further involved (author, reviewer). 

Editing

there is a graphical abstract, but not yet a 'written' abstract ? I assume that this still has to be added ? In the figure, i would mention (as done in the text) igG. 

I understand that a narrative approach has been applied, but how have topics and references been retrieved/searched. 

Is it correct to use 'neonatal lupus', while a relevant part of the paper relates to fetal conditions. I would suggest to consider perinatal, or fetal and neonatal lupus ? 

Line 95: can you support this statement with a reference ? 

Line 210: that disrupts the conduction of cardiac tissue, or that disrupts cardiac conduction tissue ? 

Line 363: suggest to add the lower mentioned references also overhere

On the use of hydroxychloroquine: there is likely some add on benefit to mention QTc prolongation (maternal, so likely also fetal ?). 

The text section on pacemaker is somewhat short and perhaps not yet sufficiently developed. How will pacing postnatal be performed (external temporary in ? hydrops/very small bw, how and when to go for final internal, technical aspects related). Either the authors have to further elaborate on this, or alternatively, make it clearer that this is specific technical topic not covered ? 

Author Response

REVIEWER 1

As reflected in the title, this paper is a narrative review on both maternal and perinatal (fetal and neonatal) lupus. I'm not sure to what extent the current title covers sufficiently well the maternal aspects as discussed. I hereby noticed that - at present - here are not obstetricians involved in the authorship, and as being a neonatologist myself i cannot fully assess the maternal aspects. Although all maternal management statements read very reasonable, perhaps obstetrics should be further involved (author, reviewer).

Dear Reviewer, Thank you for your comments and attention to our work. We would like to clarify that one of our authors, Dr. Stefano Di Michele, is part of the Division of Obstetrics and Gynecology. However, the primary focus of our study is on pediatric and neonatological aspects. The obstetric content was included mainly to provide a comprehensive clinical context, though it is not the core focus of our analysis. We believe that Dr. Di Michele’s expertise ensures adequate coverage of the obstetric aspects discussed in our paper.

There is a graphical abstract, but not yet a 'written' abstract ? I assume that this still has to be added ? In the figure, i would mention (as done in the text) igG. Thank you for your comment, a written abstract has been added.

I understand that a narrative approach has been applied, but how have topics and references been retrieved/searched. Thank you for your comment, a search strategy section has been added.

Is it correct to use 'neonatal lupus', while a relevant part of the paper relates to fetal conditions. I would suggest to consider perinatal, or fetal and neonatal lupus ? Thank you for your comment, the request change has been made.

Line 95: can you support this statement with a reference ? Thank you for your comment, the reference has been added.

Line 210: that disrupts the conduction of cardiac tissue, or that disrupts cardiac conduction tissue ? Thank you for your comment, the request change has been made.

Line 363: suggest to add the lower mentioned references also overhere. Thank you for your comment, the request change has been made.

On the use of hydroxychloroquine: there is likely some add on benefit to mention QTc prolongation (maternal, so likely also fetal ?).  Thank you for your comment, additional information related to the effects of HCQ on QTc has been added. 

The text section on pacemaker is somewhat short and perhaps not yet sufficiently developed. How will pacing postnatal be performed (external temporary in ? hydrops/very small bw, how and when to go for final internal, technical aspects related). Either the authors have to further elaborate on this, or alternatively, make it clearer that this is specific technical topic not covered? Thank you for your comment, some additional info related to postnatal pacing has been added.

Reviewer 2 Report

Comments and Suggestions for Authors

This is a very well written review on a rare neonatal condition. Overall, this review should be published to again bring light into autoimmune conditions in the neonatal period. The following are suggested to further improve this paper:

1. Consider re-arranging the sub-topics into: (1) introduction, (2) molecular mechanism, (3) congenital heart block and other complications, (4) management.

2. Please categorize the topics under molecular mechanism. Since this is the main focus of the review, the authors should organize this sub-heading into categories such as: role of cytokines, role of IgG/autoantibodies, etc., similar to how the congenital heart block sub-heading is categorized.

3. Please add placental histopathology seen in this condition.

4. Please define acronyms on first use.

Overall, this is a strong review and after the corrections mentioned above, this should warrant expedited publication.

Comments on the Quality of English Language

Excellent use of English language. However, minor grammatical, spelling and capitalization errors must be corrected.

Author Response

REVIEWER 2

This is a very well written review on a rare neonatal condition. Overall, this review should be published to again bring light into autoimmune conditions in the neonatal period. The following are suggested to further improve this paper:

  1. Consider re-arranging the sub-topics into: (1) introduction, (2) molecular mechanism, (3) congenital heart block and other complications, (4) management. Thank you for your comment, the request change has been made.

  1. Please categorize the topics under molecular mechanism. Since this is the main focus of the review, the authors should organize this sub-heading into categories such as: role of cytokines, role of IgG/autoantibodies, etc., similar to how the congenital heart block sub-heading is categorized. Thank you for your comment, the request change has been made.

  1. Please add placental histopathology seen in this condition. Thank you for your comment, the request change has been made.

  1. Please define acronyms on first use. Thank you for your comment, the request change has been made.

Overall, this is a strong review and after the corrections mentioned above, this should warrant expedited publication.